# Evaluating the impacts of microplastics on agricultural soil physical, chemical properties, and toxic metal availability: An emerging concern for sustainable agriculture

Tapos Kumar Chakraborty[1], Md. Simoon Nice[1]*, Md. Sozibur Rahman[1], Baytune Nahar Netema[1], Khandakar Rashedul Islam[1], Samina Zaman[1], Gopal Chandra Ghosh[1], Md. Ripon Hossain[1], Asadullah Munna[1], Mst. Shamima Akter[1], Md. Abu Rayhan[1], Sk Mahmudul Hasan Asif[1], Abu Shamim Khan[2]

1 Department of Environmental Science and Technology, Jashore University of Science and Technology, Jashore, Bangladesh, 2 Environmental Laboratory, Asia Arsenic Network, Jashore, Bangladesh

* simoon.nice.est.just@gmail.com

**Data Availability Statement:** All relevant data are within the manuscript and its Supporting Information files.

## Abstract

Microplastics (MPs) are an emerging environmental issue that might endanger the health of agricultural soil. Even though several research on the particular toxicity of MPs to species have been carried out, there is little information on MPs' impacts on soil physicochemical properties and heavy metals (HMs) availability of HMs contaminated and without contaminated soils. This study examined the changes in soil characteristics for both HMs contaminated and without contaminated soils by five distinct MPs, including Polyethylene (PE), Polyethylene terephthalate (PET), Polystyrene Foam (PS), Polyamide (PA), and a combination of these four types of MPs (Mixed MPs), at two different concentrations (0.2% and 1%; w/w), where soil incubation experiments were setup for this studies and the standard analytical techniques employed to measure soil characteristics and toxic metal availability. After the ending of soil incubation studies (90 days), significant changes have been observed for physicochemical properties [bulk density, porosity, water holding capacity, pH, electrical conductivity (EC), organic carbon (OC), and organic matter (OM)]. The soil nutrients change in descending order was found as $NH_4^+$-N> $PO_4^{3+}$ > Na > Ca > $NO_3^-$ > Mg for lower concentrations of MPs compared to higher concentrations. The HMs availability is reducing with increasing MPs concentration and the descending order for metal availability was as follows Pb > Zn > Cd > Cr > Cu > Ni. Based on MP type, the following descending order of MPs PS > Mix (MPs) > PA > PET > PE, respectively act as a soil properties influencer. Usually, effects were reliant on MPs' category and concentrations. Finally, this study concludes that MPs may modify metal movements, and soil quality; consequently, a possible threat will be created for soil health.

**Funding:** The author(s) received no specific funding for this work.

**Competing interests:** The authors have declared that no competing interests exist.

## 1. Introduction

Plastics, the synthetic polymers that are used largely indiscriminately in the modern world due to their low production cost, lightweight, elasticity, and durability properties [1]. Since 1950, worldwide plastic production dramatically increased, and the predicted total manufactured plastics was 348 million metric tonnes in 2050 with an annual increase of 33 billion tonnes globally [2]. Polyethylene (PE), Polystyrene (PS), Polypropylene (PP), Polyethylene Terephthalate (PET), and Polyvinylchloride (PVC) are the most commonly produced and used plastic polymers [3]. Due to durability, unsustainable use, inadequate waste management, and low recycling rates, plastics tend to accumulate substantially in natural ecosystems [4]. Around 20% of plastics are recycled, whereas the existing 80% are ultimately gathered in different environmental matrices such as soil and water bodies [5]. The manufactured plastic particles or the breakdown of larger plastic items introduces an emerging contaminant into the environment called "MPs" (longest dimension <5mm) [6] have received scientific concern due to their pollution and risk into the soil, air, and water ecosystem. Since the majority of plastic debris is generated and released on land, soils can act as a significant long-term sink for MP particles [7]. Urban and farmland soil are thought to be susceptible to MP emissions because they constitute the hub of human activity and, hence the channels for MP input [8]. According to reports, MPs had contaminated 90% of the soil in the Swiss floodplains [9]. Numerous studies revealed MPs in soil, particularly in farmland, 12–117 items/m$^2$ [8], 78.00±12.91 items/kg [10], $4.3×10^4$ to $6.2×10^5$ particles/kg [11]. MPs may enter the soil from a variety of sources, including compost, wastewater irrigation, sludge, plastic mulching, surface runoff, and atmospheric deposition [12]. As a result, MPs in farmland soil can adversely affect the soil ecosystem [13]. When entering the soil, MPs will change soil physiochemical properties [12], ecosystem functioning, and microbial population either directly or indirectly [14]. Several studies reported that the soil pH [15], soil structure [16], and soil fertility [17] can be changed by MPs. The effects of MPs on plant growth, heavy metals uptake, crop yields, soil health, and management are carried out by several studies [18–23]. Therefore, the buildup of MPs in farmland soil can negatively impact the health and functioning of soil ecosystems and ultimately pose hazards to the safety of the food chain [24]. According to Holmes [25] and Rillig [26], this new environmental anthropogenic stressor not only directly harms soil organisms but also has the potential to synergistically pollute the environment with other pollutants like heavy metals. Recent research suggests that because of their tiny size, high hydrophobicity, and greater surface area-to-volume ratio, MPs are capable of absorbing organic pollutants and heavy metals on their surfaces under a variety of environmental conditions [11, 25]. Soil health and agroecosystems have potential threats due to the exposure of MPs and heavy metals (HMs) [11, 27]. Various research found that MPs in agroecosystems could alter the bio-availability and characteristics of HMs (As, Cd, Cr, Ni, Cu, Pb, and Zn) [28]. For example, MPs enhance Cd toxicity in earthworms by increasing bioavailability [29]. In addition, MPs and soil contaminants like Cd and nano-ZnO alter symbiotic fungi and plant growth due to MPs enhancing the availability of these elements, thereby threatening soil biodiversity and agroecosystems [30]. However, there are significant knowledge gaps regarding the relationship between changes in soil properties under MPs co-occurring with other heavy metals for understanding the ecological effect on soil [12]. Numerous studies have reported the MPs abundance in freshwater or marine ecosystems [10]. According to Andrady [6] and Horton [31] the majority of the plastic garbage in fresh and marine water environments is derived from land-based sources. MPs and toxic metals in agricultural soil together alter the soil's physical and chemical properties, and structure of the biological community even plant growth, and production rate [15, 30]. Additionally, MPs are also able to enhance the bioavailability rate of toxic elements in agricultural soil [12].

Sa'adu and Farsang [32] and Fakour et al. [8] broadly review the existence of plastic in agricultural soil, where most of the study focused on the occurrences, distribution, and characterization of MPs, only a few studies were conducted on the MPs impacts on agricultural soil properties [12, 33]. So, there is a big knowledge gap on the impacts of agricultural soil properties by MPs and toxic metals jointly, which is a matter of great concern for agricultural practices, and ecological risks [34]. Therefore, the objective of this study was to assess the impacts of MPs on soil physical, and chemical properties and toxic metals availability in contaminated and without contaminated agricultural soils.

## 2. Materials and methods

### 2.1. Microplastic preparation

In this study, different types of new plastic products were used as the sources of different types of microplastic. Based on previous literature such as Feng et al. [12], Wen et al. [35], and Li et al. [36], this study chooses the following plastic polymers including Polyethylene terephthalate (PET), Polyethylene (PE), Polystyrene Foam (PS) and Polyamide (PA) respectively from the using of plastic bottle, polyethylene paper, foam and nylon. Finally, these polymers are also confirmed by ATR-FTIR. Initially, plastic products were cut into smaller pieces, passed through a 5mm sieve to ensure the desired size, then washed initially with tap water, and secondly washed with double-distilled water. Finally, it was cleaned with 0.1 N $HNO_3$ to eliminate any remaining organic material from their surfaces. After washing, all of these particles were dried at 60°C in a Labtech LDO-150F oven (Korea), cooled to room temperature, and ready to use for study purposes.

**2.1.1. Soil preparation and experiment set-up.** The test soil is a sandy clay loam soil (sand 57%, silt 22%, clay 21%) taken from nearby vegetable farmland (by taking the permission from the land owner), located at Abdulpur Village, Jashore District, Bangladesh (23° 13' 43.23" N, 89° 8' 53.14" E), where USDA soil taxonomy methods were applied form soil type identification [37]. The soil was collected from the top layer (0–20 cm) of the above-mentioned farmland. The soil was air-dried, grounded, and passed through a 2 mm sieve for soil incubation experiments. Finally, two different types of groups were created, normal soil (without contamination) and other is artificially contaminated soil with heavy metals (contaminated) the metals salt solution into the soil and then kept the soil in the dark for 7 days to complete the metal doping, this method was adopted from Feng et al. [12]. After 7 days the soil was dried, ground, and sieved as explained previously. This study conducted a three-factorial pot experiment on 10 August 2022. Four different types of MPs were used for this test including Polyethylene terephthalate (PET), Polyethylene (PE), Polystyrene Foam (PS), and Polyamide (PA), lastly, an equal portion of the above-mentioned MPs were mixed and defined as Mixed MPs. Therefore, five different categories of MPs (eg. PE, PET, PS, PA, and Mix MPs), were added to both normal (without contamination), and artificially contaminated soil at two different concentrations 0.2% and 1%, w/w selected based on previous literature Feng et al. [12], and Li et al. [36] while a controlled study (without adding MPs) also conducted at the same time for both normal and contaminated soil. A total of 22 pots (11 without contaminated and 11 for contaminated soil) were filled with 200g soil-MP mixtures. Throughout the whole incubation, soil moisture was kept at 30% of its maximal water-holding capacity. To prevent water evaporation, Parafilm® was used to seal the cup's (clay-made cup) top, shown in S1 Fig, adopted from Feng et al [12]. All the cups were placed randomly and kept in darkness at 25 ± 0.5°C. Following 90 days of incubation, the soil was taken for chemical analysis. Triplicate experiments were conducted using a similar procedure and finally average value was used for experiment results.

## 2.2. Analysis of soil properties

Analytical-grade chemicals and reagents were used throughout the investigation. In a soil water suspension (soil-water ratio 1:2.5), soil pH was determined by using a pH meter (Milwaukee pH56 Martini Pocket pH meter, Romania), while EC was assessed by using an EC meter (HACH Sension-156; multi-parameter, USA). The contents of porosity, bulk density, and water holding capacity analysis protocol were adopted by Carter and Gregorich [38]. For $PO_4^{3-}$ (Detection ranges: 0.02 to 2.50 mg/L) determination; the soil extraction process was adopted from Mussa et al. [39] and measured using an ultraviolet spectrophotometer (HACH DR 3900) at 880 nm. 2 mol/L KCl was used to extract the nitrogen from the soil, and an ultraviolet spectrophotometer was used to measure the amounts of $NH_4^+$ (Detection ranges: 0.4 to 50.0 mg/L) and $NO_3^-$ (Detection ranges: 0.1 to 10.0 mg/L). The extracted solution was evaluated at 400 nm for the analysis of $NO_3^-$, and at 655 nm for the measurement of $NH_4^+$ [12]. Soil OC and OM were determined by using the Walkley–Black titrimetric method [37]. Following Allen et al. [40] instructions soil samples were digested using the tri-acid combination [$HNO_3$ (69%): $H_2SO_4$ (98%): $HClO_4$ (70%) = 5:1:1]. A 250 mL conical flask was filled with precisely 1.00 g of crushed material, which was then digested with 15 mL of the tri-acid solution at 180–200°C until a clear solution was obtained. The digested solution was then cooled to a temperature of around 25°C, filtered using Whatman 41 paper, and diluted to a volume of 100 mL with double-distilled water. Blank samples were also prepared using a similar process. Graphite furnace, hydride initiator, and air-acetylene flame Atomic-absorption-spectrophotometer (AAS) (Model: AA-7000, SHIMADZU, Japan) were used to measure the concentration of Cd, Cr, Pb, Ni, Cu, Zn, Na, Ca, and Mg in the samples at a specific wavelength. The determination limit of Cd was 0.004 mg/L, while the level of determination of Na, Mg, and Ca was 0.01–0.004 mg/L and ranges between 0.013–0.070 mg/L for HMs (Ni, Pb, Cr, Cu, and Zn). A diverse concentration of working solution was prepared from the standard solution (1,000 ppm, Sigma Aldrich, USA) for calibration curve development. The output of analysis results was stated as mg/L, then converted into mg/kg. Double-deionized water was used for all laboratory experiments. All glassware and instruments were properly cleaned before use. For better quality assurance, bank sample runs and duplicate analyses were made for each sample, analysis samples were diluted based on needs. DORM-4 Fish Protein (NRC, CANADA) was used as certified reference materials, where the deviation of recovery rate for measured elements was within ± 5–7%.

## 2.3. Statistical analysis

SPSS V.16.0 (SPSS, USA) was used for ANOVA analysis, where significance level was selected as $p < 0.05$. Microsoft Office LTSC Professional Plus 2021 used graph preparation, and data calculation. Calculations were made about the availability of heavy metal contents in soil samples as well as the available soil property contents.

## 3. Results and discussion

### 3.1. Effects of MPs on soil physicochemical properties

pH is considered as a most vital parameter in soil because optimum pH enhances essential nutrients for plants. The level of pH in the soil is significantly affected by MPs abundance, types, polymers, and duration of incubation [41]. This study result shows that the PS increases soil pH with increasing MP concentration (0.2–1%) for contaminated (5.97–6.85) and without contaminated soil (6.42–6.93) due to increases of soil aeration and porosity or leaching of chemical additives [42]. The surface properties of MPs might be insisting on adsorbing diverse

cations and anion-charging elements, which alter the soil pH [12]. Qi et al. [43] found that MPs increase soil pH. Conversely, other MPs (PE, PET, PA, and mixed) reduce the soil pH for both contaminated and without contaminated soil and might release organic acid from MPs through mineralization [44]. Feng et al. [12] reveal that PE, PS, PA, and PBS MPs reduce the soil pH. Electrical conductivity (EC) is very vital for soil health, elevated and lower levels of EC reduce nutrient availability and accessibility for plants. MPs types and abundance can change the EC value of soil [45]. This study shows that PE, PS, PA, and Mix (MPs) increase the soil EC value at lower concentrations (0.2%, w/w) but decrease at higher concentrations (1%, w/w) in both contaminated and without contaminated soil (Table 1). This table shows that the lower concentration of PE is increasing the EC value of the soil at a significant rate in both instances. Additionally, in terms of PET, the soil EC value is decreasing with the increase of concentration for both contaminated and without contaminated soil (Table 1). MPs alter soil porosity and bulk density, which are two contributing elements that influence soil EC [42]. Soil organic carbon and organic matter significantly influence soil fertility, which reduces soil erosion and nutrient leaching rates parallel increasing soil aeration, water drainage, and retention. Experimental results show that all MPs types (PE, PET, PS, PA, and mixed) change the contents of OC and OM percentages in contaminated (OC = 0.468–0.265%, and OM = 1.001–0.840%) and without contaminated (OC = 0.446–0.375%, and OM = 0.105–0.593%) soil at low concentration (0.20%) than higher concentration (1%). More specifically, mixed MPs show the highest reduction rate for both soils (Table 1). MPs alter the structure of microorganism community and their activity which influences the decomposition and conversion of organic materials [17]. Zhang et al. [46] reveal that MPs gathering in agricultural soil could cover soil organic carbon storage. Cao et al. [47], and Liu et al. [17] explore that MPs have significant negative effects on SOC and SOM, respectively. Dong et al. [48] found that Polytetrafluorethylene and Polystyrene reduce by about 34.3% and 25.8%, SOM, respectively. Soil bulk density represents the soil's capacity to support structures, convey water and solutes, and aerate the soil. MPs present in soil influence the change in bulk density. This study found bulk density decreases at a larger rate at lower concentrations than at higher concentrations for contaminated ($1.3–1.03$ g/cm$^3$) and without contaminated soil ($1.25–1.01$ g/cm3) (Table 1). A study by Qi et al. [43] also found a similar kind of change in the presence of MPs. Soil porosity enhances the availability and mobility of water and air within the soil environment. The percentage of pore space in experimental soil gradually reduces with increasing MPs concentration for both contaminated (48–32%) and without contaminated soil (50–34%) through reducing the soil pore numbers and size [49] among all MPs, PS shows the highest effect. de Souza Machado et al. [50] also discovered similar types of changes. Soil water holding (WHC) capacity defines the soil productivity. Soil with high water holding capacity improves the crop yield. However, the presence of MPs changes the WHC for both contaminated (75–59%) and without contaminated soil (80–62%) (Table 1) due to altering the soil texture and organic matter. This study has found that the water-holding capacity is decreasing at a significant rate with the change in MP concentrations [41]. Wang et al. [51] found that high concentration of MPs reduces the soil water holding capacity.

## 3.2. Effects of MPs on soil nutrient availability

Soil micro and macro nutrients maintain soil fertility and it's highly needed for plant growth, development, and production. Too much and too low nutrients reduce soil quality and agricultural production [51]. The descending order of MPs for soil nutrients (NO3, PO4, NH4, Na, Ca, and Mg) are mixed MPs > PET > PS > PA > PE and mixed MPs > PS > PA > PET > PE for contaminated and without contaminated soil, respectively. Sodium (Na) helps to

**Table 1. Effects of MPs on soil physicochemical parameters in both contaminated and without contaminated soil.**

| MPs Type | MPs Concentration (w/w) | pH | | EC (mS/cm) | | OC (%) | | OM (%) | |
|---|---|---|---|---|---|---|---|---|---|
| | | Without Contaminated Soil[a] | Contaminated Soil[b] | Without Contaminated Soil[c] | Contaminated Soil[d] | Without Contaminated Soil[e] | Contaminated Soil[f] | Without Contaminated Soil[g] | Contaminated Soil[h] |
| | Control | 6.42 | 5.97 | 0.4432 | 0.4432 | 0.531 | 0.531 | 1.191 | 1.191 |
| PE | 0.20% | 6.31 | 5.9 | 0.5733 | 0.5733 | 0.469 | 0.447 | 1.051 | 1.001 |
| | 1% | 6.21 | 5.92 | 0.4195 | 0.4195 | 0.431 | 0.5 | 0.966 | 1.121 |
| PET | 0.20% | 6.14 | 5.8 | 0.4276 | 0.4276 | 0.391 | 0.453 | 0.875 | 1.015 |
| | 1% | 6.08 | 5.9 | 0.4193 | 0.4193 | 0.469 | 0.406 | 1.051 | 0.91 |
| PS | 0.20% | 6.76 | 6.22 | 0.476 | 0.476 | 0.447 | 0.469 | 1.001 | 1.051 |
| | 1% | 7.03 | 6.85 | 0.4035 | 0.4035 | 0.422 | 0.517 | 0.945 | 1.158 |
| PA | 0.20% | 6.1 | 5.6 | 0.4718 | 0.4718 | 0.406 | 0.5 | 0.91 | 1.121 |
| | 1% | 6.33 | 5.91 | 0.3998 | 0.3998 | 0.484 | 0.203 | 1.085 | 0.455 |
| Mix | 0.20% | 6.4 | 5.8 | 0.4477 | 0.4477 | 0.265 | 0.375 | 0.594 | 0.84 |
| | 1% | 6.21 | 5.86 | 0.4117 | 0.4117 | 0.432 | 0.469 | 0.968 | 1.051 |

| MPs Type | MPs Concentration (w/w) | Bulk Density (g/mL) | | Porosity (%) | | Water Absorption Capacity (%) | | | |
|---|---|---|---|---|---|---|---|---|---|
| | | Without Contaminated Soil[i] | Contaminated Soil[j] | Without Contaminated Soil[k] | Contaminated Soil[l] | Without Contaminated Soil[m] | Contaminated Soil[n] | | |
| | Control | 1.25 | 1.3 | 50 | 48 | 80 | 75 | | |
| PE | 0.20% | 1.063 | 1.07 | 47.37 | 47.03 | 77 | 72 | | |
| | 1% | 1.122 | 1.15 | 44.44 | 43.07 | 72 | 69 | | |
| PET | 0.20% | 1.092 | 1.099 | 49.91 | 49.59 | 68 | 64 | | |
| | 1% | 1.263 | 1.25 | 43.75 | 42.66 | 60 | 59 | | |
| PS | 0.20% | 1.01 | 1.03 | 40 | 38.81 | 65 | 62 | | |
| | 1% | 1.11 | 1.13 | 34.07 | 32.87 | 59 | 57 | | |
| PA | 0.20% | 1.122 | 1.14 | 48.99 | 48.18 | 73 | 67 | | |
| | 1% | 1.195 | 1.2 | 46.75 | 45.45 | 65 | 62 | | |
| Mix | 0.20% | 1.015 | 1.03 | 44.72 | 43.91 | 70 | 66 | | |
| | 1% | 1.13 | 1.17 | 38.47 | 36.29 | 62 | 60 | | |

ANOVA

([a] $F = 1.07$, $P>0.05$

[b] $F = 1.08$, $P>0.05$

[c] $F = 1.24$, $P<0.05$

[d] $F = 7.28$, $P<0.05$

[e] $F = 1.91$, $P<0.05$

[f] $F = 0.24$, $P<0.05$

[g] $F = 1.92$, $P>0.05$

[h] $F = 0.23$, $P>0.05$

[i] $F = 2.826$, $P<0.05$

[j] $F = 12.76$, $P>0.05$

[k] $F = 2.81$, $P>0.05$

[l] $F = 3.20$, $P>0.05$

[m] $F = 5.04$, $P>0.05$

[n] $F = 3.24$, $P>0.05$)

keep the soil fertile by marinating basal performance and also helps plants utilize water efficiently by controlling the osmotic pressure of the cells [52, 53]. The experimental study shows that initial concentrations of MPs (0.20%) decrease the Na contents from contaminated (1–

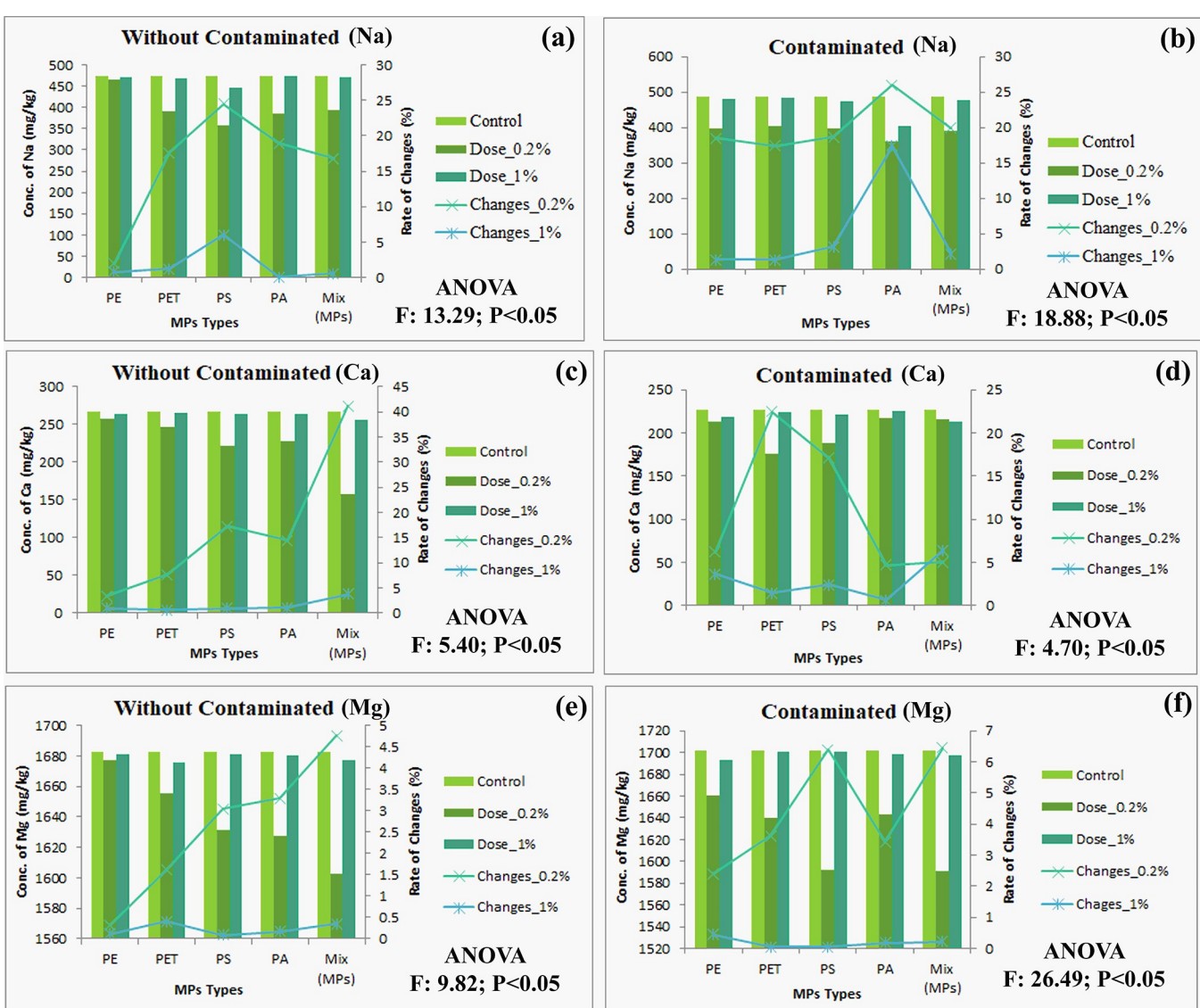

**Fig 1.** Effects of MPs on soil nutrients availability in both without contaminated and contaminated soil, (a-b) Na; (c-d) Ca; (e-f) Mg; respectively, significant at P<0.05.

24%) and without contaminated soil (17–26%) (Fig 1A and 1B). Calcium (Ca) regulates cell wall structure and membranes additionally it plays a vital role in balancing organic acid and enzyme systems [54]. The concentration of Ca in soil significantly reduces in both soil (contaminated = 4–22%, and without contaminated = 3–41%) (Fig 1C and 1D). Magnesium (mg) assists plant physiology and biochemical activities, it acts as a key element for plant growth, and development and protecting agent for reducing abiotic stress [55]. The abundance of MPs in agricultural soil slightly declines the Mg contents (Contaminated = 2–7%, and without contaminated = 0.3–4.76%) (Fig 1E and 1F). MPs directly and indirectly alter the soil physicochemical properties (pH, temperature, moisture, OC, OM, structure, texture, etc.) and microbial activities, which enhance soil permeability, nutrient leaching rate, and availability of elemental concentration [31]. Plants uptake inorganic nitrogen from the soil, mostly as $NH_4^+$ and $NO_3^-$ that significantly stimulate plant growth [56]. The concentration of $NH_4^+$ (Contaminated = 8–50%, and without contaminated = 8–48%) (Fig 2E and 2F), and $NO_3^-$

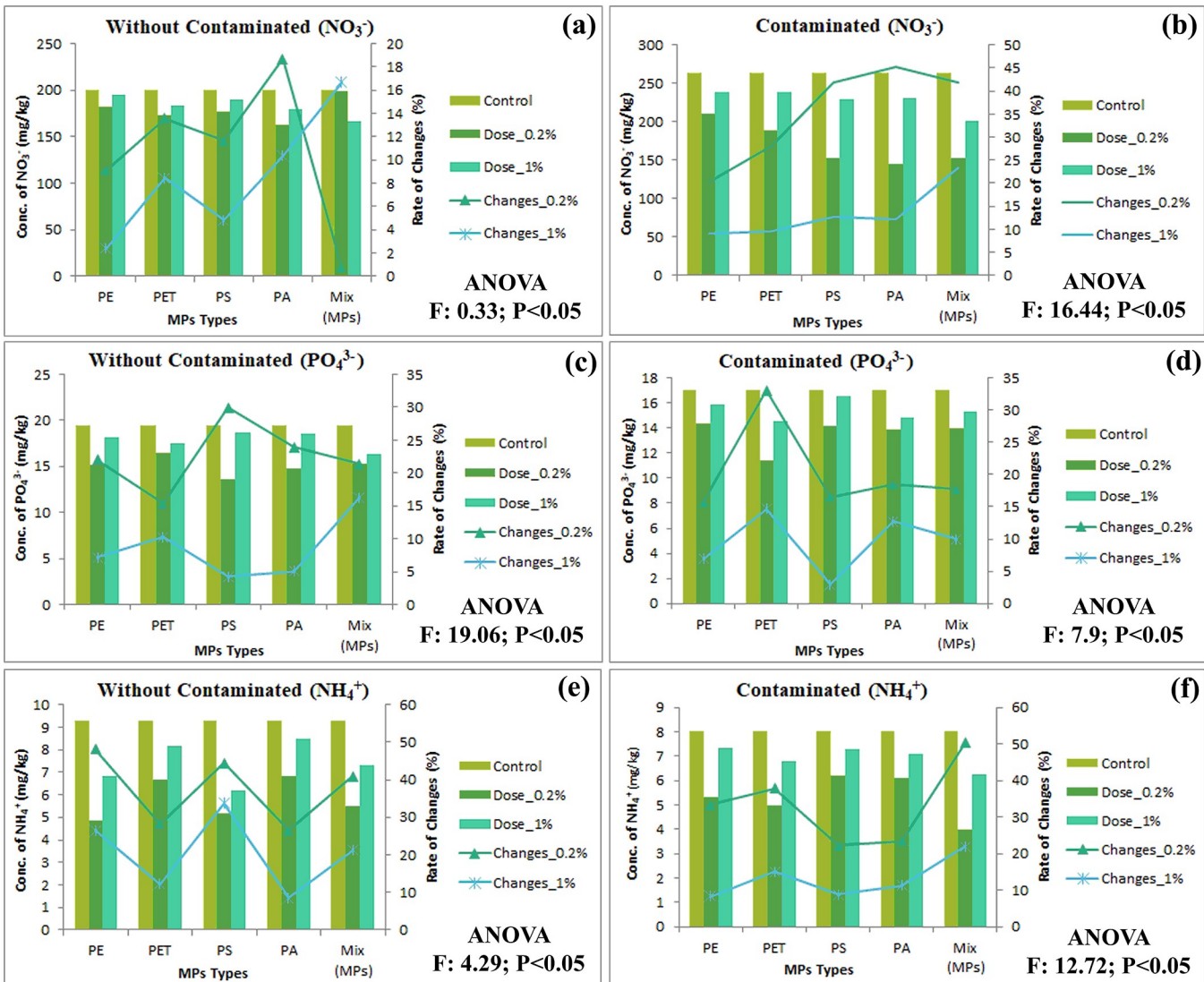

**Fig 2.** Effects of MPs on soil nutrients availability in both without contaminated and contaminated soil, (a-b) $NO_3^-$; (c-d) $PO_4^{3-}$; (e-f) $NH_4^+$; respectively, significant at $P<0.05$.

(Contaminated = 9–45%, and without contaminated = 0.72–19%) (Fig 2A and 2B) are reduces from soil by MPs due to leaching from agro-ecosystems, altering the soil surface functional groups, and hindering the actions of main enzymes in the soil nitrogen cycle [57]. Zhu et al. [58] found that MPs reduced $NO_3^-$N concentration by up to 91%. The contents of phosphate in soil are reduced (without Contaminated = 17–11.39 mg/kg, and contaminated = 19.5–13.67 mg/kg) (Fig 2C and 2D), which might be constraining soil enzyme actions. Li and Liu [59] exhibited that the concentration of phosphate in soil was reduced from 122.61 mg/kg to 63.43 mg/kg by MPs. This remarkable change of nutrients in agricultural soil reduces soil fertility, crop yield, and soil health.

### 3.3. Effects of MPs on soil metals availability

MPs act as a vector that carries potentially hazardous elements (eg. heavy metals) from the surrounding environment; consequently, it declines the soil quality by triggering the synergistic

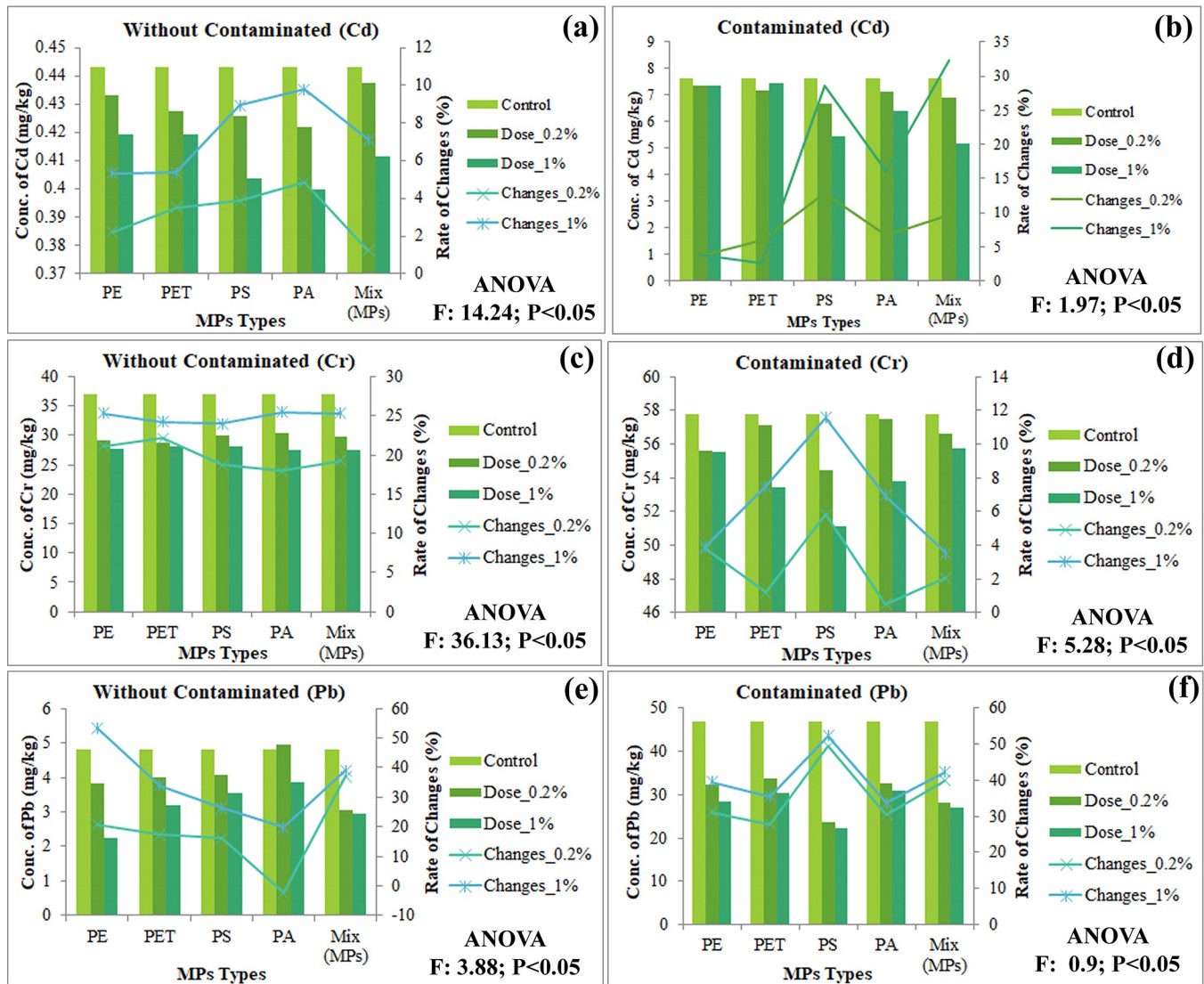

**Fig 3.** Effects of MPs on soil metals availability in both without contaminated and contaminated soil, (a-b) Cd; (c-d) Cr; (e-f) Pb; respectively, significant at P<0.05.

effect of MPs-HMs [60] where MPs influence the relocation and alteration of HMs through adsorption, precipitation or modifying the physiochemical parameters of soil [28]. This study's findings show that MPs significantly adsorb the HMs from the experimental soil and the descending order of HMs were Pb > Zn > Cd > Cr > Cu > Ni (Figs 3 & 4), where S2 and S3 Figs confirmed that all forms of MPs are involved. The distinct properties of MPs including small particle size, bulky surface area, lipophilic nature, and specific morphological features directly involved reducing the bioavailability of HMs, while MPs indirectly decrease HMs availability by altering the soil properties such as physical, chemical, and biological properties [28, 60]. Yuan et al. [61] found that MPs reduce HMs availability and the following order was Pb > Cu > Cd > Ni. Feng et al. [12] also found that MPs significantly adsorb Zn and Pb from agricultural soil. Yu et al. [62, 63] found that direct adsorption and indirect modification of soil microenvironment reduces the HMs bioavailability by polyethylene MPs. Besides, the

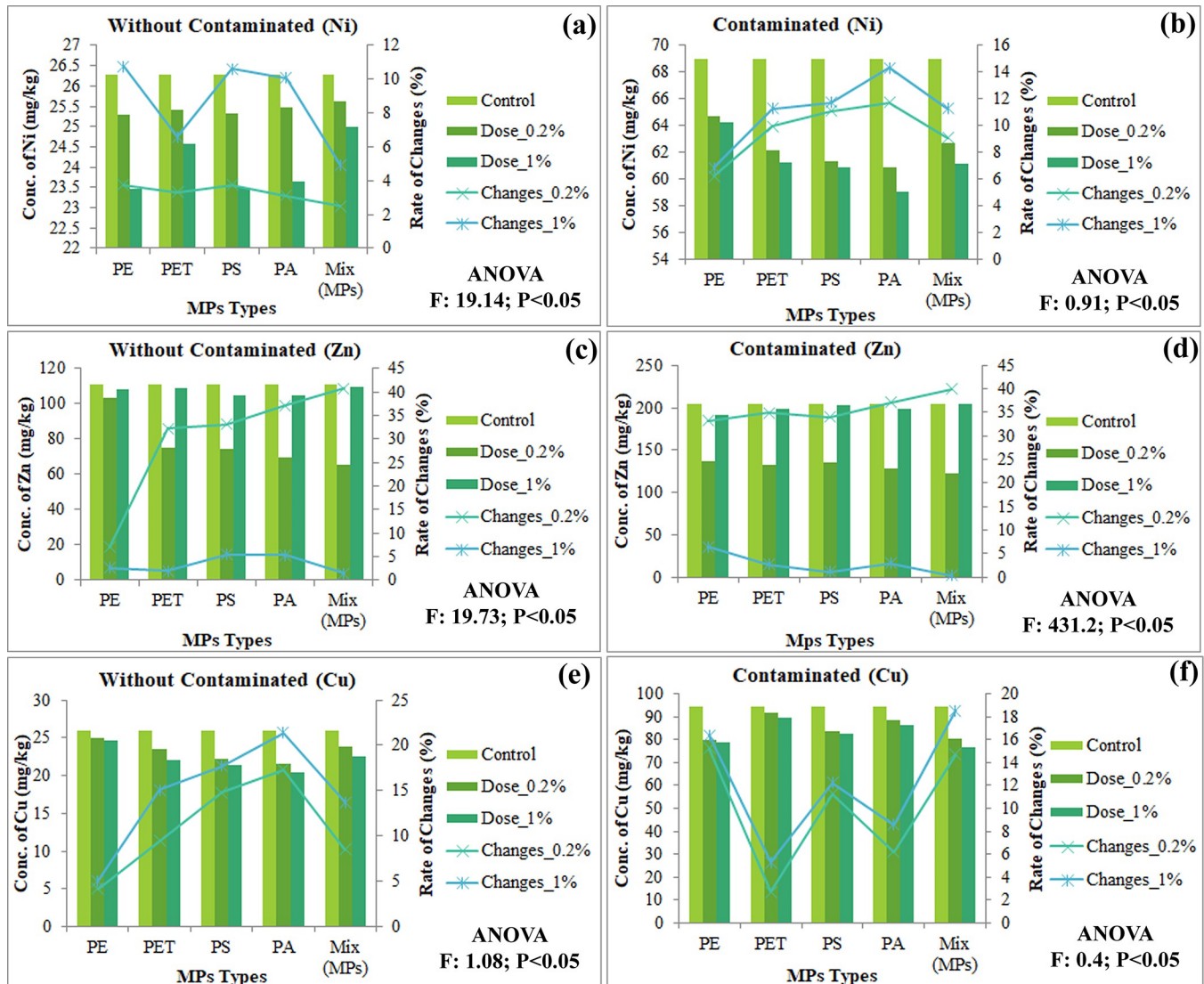

**Fig 4.** Effects of MPs on soil metals availability in both without contaminated and contaminated soil, (a-b) Ni; (c-d) Zn; (e-f) Cu; respectively, significant at P<0.05.

bioavailability of HMs also regulated by diverse factors including pH, cation exchange capacity, oxidation-reduction potential, percentage of organic matter, and microbial action [64, 65]. Lastly, this study shows that MP types and doses may have a significant for HMs availability.

## 4. Limitations of the study

This study was conducted based on four types of MPs (size< 5mm) where other MP categories and MP size variation were not considered. In the whole experiment, only one soil type (sandy clay loam) was selected no variations were considered. Effects of MPs on plant response were not carried out due to time shortages, and lack of facilities. This study only observed the changes in soil properties by adding MPs, but the mechanisms behind these changes and why specific MPs are responsible for these changes were not evaluated. Here, all discussed limitations should be overcome for our future studies.

## 5. Conclusions

After a 90-day soil incubation experiment, this study found that soil chemical properties (pH, EC, OC, OM) and physical properties (bulk density, porosity water absorption capacity) are changing significantly with concentration variation, besides soil nutrient (Na, Ca, Mg, $NO_3^-$, $PO_4^{3-}$, $NH_4^+$-N) availability are mostly decreasing more at lower concentration while higher concentration poses negligible changes. The bioavailability of heavy metals (Cd, Cr, Pb, Ni, Zn, and Cu) decreases with the concentration increases, except for Zn. Typically, all the effects varied with MP type and applied concentration. Except for soil nutrients, higher MP concentrations largely exhibited a substantial influence. Further studies are necessary to investigate the mechanisms underlying the interaction between MPs with HMs, and possible effects on other physicochemical parameters, living organisms, and soil health. Plant experiments are recommended for further studies to understand the joint effects of MPs and HMs on plant growth in a real environment.

## Supporting information

**S1 Fig. Pot-experiment set up.**
(TIF)

**S2 Fig.** EDS of MPs after experiments (a) PET, (b), PS, (c), PE, (d) PA, (e) Mixed MPs.
(TIF)

**S3 Fig.** SEM of MPs after experiments for before (b) PET, (d), PS, (f), PE, (h) PA, (j) Mixed MPs and after (a) PET, (c), PS, (e), PE, (g) PA, (i) Mixed MPs.
(TIF)

**S1 Data.**
(XLSX)

## Acknowledgments

The authors would like to thank the Department of Environmental Science and Technology, Jashore University of Science and Technology, Jashore 7408, Bangladesh for laboratory facilities.

## Author Contributions

**Conceptualization:** Tapos Kumar Chakraborty, Md. Simoon Nice, Samina Zaman, Gopal Chandra Ghosh.

**Data curation:** Md. Sozibur Rahman, Baytune Nahar Netema, Khandakar Rashedul Islam, Mst. Shamima Akter, Md. Abu Rayhan, Sk Mahmudul Hasan Asif.

**Formal analysis:** Md. Sozibur Rahman, Baytune Nahar Netema, Khandakar Rashedul Islam, Mst. Shamima Akter, Md. Abu Rayhan, Sk Mahmudul Hasan Asif.

**Investigation:** Tapos Kumar Chakraborty, Samina Zaman, Gopal Chandra Ghosh, Md. Ripon Hossain, Asadullah Munna.

**Methodology:** Tapos Kumar Chakraborty, Samina Zaman, Gopal Chandra Ghosh, Md. Ripon Hossain, Asadullah Munna.

**Software:** Gopal Chandra Ghosh, Abu Shamim Khan.

**Writing – original draft:** Tapos Kumar Chakraborty, Md. Simoon Nice, Samina Zaman, Gopal Chandra Ghosh.

**Writing – review & editing:** Tapos Kumar Chakraborty, Samina Zaman, Mst. Shamima Akter, Md. Abu Rayhan, Sk Mahmudul Hasan Asif.

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
