## [Decision Letter · Decision Letter 0]

9 Apr 2024

PONE-D-23-41961Short Communication: Evaluating the impacts of microplastics on agricultural soil physicochemical properties and toxic metal availability: an emerging concern for sustainable agriculturePLOS ONE

Dear Dr. Nice,

Thank you for submitting your manuscript to PLOS ONE. After careful consideration, we feel that it has merit but does not fully meet PLOS ONE’s publication criteria as it currently stands. Therefore, we invite you to submit a revised version of the manuscript that addresses the points raised during the review process.

**ACADEMIC EDITOR: Kindly revise your manuscript according to the reviewers comments and give them positive response. **Please ensure that your decision is justified on PLOS ONE’s publication criteria and not, for example, on novelty or perceived impact.

We look forward to receiving your revised manuscript.

Kind regards,

Babar Iqbal, PhD

Academic Editor

PLOS ONE

Journal Requirements:

Additional Editor Comments:

Carefully, revise your manuscript especially reviewers two and give him a positive response. Otherwise, your manuscript will be consider rejected.

Reviewers' comments:

Reviewer's Responses to Questions

**Comments to the Author**

1. Is the manuscript technically sound, and do the data support the conclusions?

Reviewer #1: Partly

Reviewer #2: Partly

2. Has the statistical analysis been performed appropriately and rigorously? 

Reviewer #1: N/A

Reviewer #2: No

3. Have the authors made all data underlying the findings in their manuscript fully available?

Reviewer #1: No

Reviewer #2: No

4. Is the manuscript presented in an intelligible fashion and written in standard English?

Reviewer #1: Yes

Reviewer #2: No

5. Review Comments to the Author

Reviewer #1: Dear authors

Abstract

The abstract effectively summarizes the main points of the study. However, some sentences could be further clarified and condensed for brevity without losing important information.

Provide more specific details about the methods used in the study, such as the experimental setup for soil incubation studies and the analytical techniques employed to measure soil characteristics and heavy metal availability.

Instead of listing all the changes in soil characteristics and heavy metal availability in a single sentence, consider breaking down the findings into separate sentences or bullet points to improve readability and comprehension.

Include specific quantitative results or ranges, such as percentage changes in soil properties and heavy metal concentrations, to provide readers with a clearer understanding of the magnitude of the observed effects.

Expand on the implications of the findings for soil health and environmental management. Discuss how the observed changes in soil properties and heavy metal availability due to MPs contamination might impact ecosystem functions and human health.

Ensure consistency in language use and grammar throughout the abstract. For example, check for proper tense usage and grammatical structure in each sentence.

The selected keywords effectively highlight the main topics of the study. However, consider adding additional keywords related to soil pollution, environmental impact assessment, or plastic pollution to improve the discoverability of the manuscript.

Provide a concise summary of the key conclusions drawn from the study's findings, emphasizing the significance of the research and potential avenues for future investigation.

.

Introduction

The Introduction effectively introduces the topic of microplastic pollution in soil and highlights its significance for soil health and ecosystem functioning. The structure of the Introduction is logical, with clear transitions between different aspects of the topic.

The Introduction provides a thorough review of relevant literature, citing key studies to support the discussion on the sources, distribution, and potential impacts of microplastics in soil ecosystems. However, ensure that all cited studies are accurately referenced and consider including more recent references to reflect the latest research in the field.

The objectives of the study are clearly stated, indicating the focus on assessing the impact of MPs on soil properties and heavy metal availability in both contaminated and uncontaminated agricultural soils. Consider briefly mentioning the specific MPs types and doses tested in the study to provide additional context for readers.

The Introduction effectively identifies knowledge gaps in the current understanding of microplastic pollution in terrestrial environments, emphasizing the need for research on MPs in agricultural soils. This sets the stage for the study's contribution to addressing this gap.

Ensure smooth transitions between paragraphs and sections to maintain the flow of the Introduction. Consider revising sentence structures for improved clarity and readability, avoiding overly complex or convoluted phrasing.

Conclude the Introduction by emphasizing the importance of the study's findings for informing soil management practices, environmental policy, and future research directions in the field of microplastic pollution in terrestrial ecosystems.

The review of intermittent irrigation techniques is good but citing key studies would strengthen the evaluation (e.g. "Recent work like doi.org/10.1002/ep.14301, https://doi.org/10.1080/15320383.2023.2258413, https://doi.org/10.1080/15226514.2023.2250464, doi.org/10.1002/ep.14230, doi.org/10.1002/ep.14217, https://doi.org/10.1007/s10333-022-00915-5

Materials and Methods" section needs significant revisions for clarity and completeness:

The section effectively outlines the steps involved in microplastic preparation, soil preparation, experimental setup, and analysis methods. However, some sentences are lengthy and complex, which may affect readability. Consider breaking down long sentences into shorter ones for clarity.

The description of microplastic preparation is clear and provides sufficient detail on the sources of different types of microplastics and the cleaning process. Ensure consistency in terminology (e.g., "plastics items" vs. "plastic products") for clarity.

The description of soil collection, preparation, and experimental setup is well-detailed, including information on soil properties and treatment groups. Consider providing more information on the rationale for choosing specific microplastic types and doses for the experiment.

The methods for analyzing soil properties, including pH, nutrient content, organic carbon, and heavy metals, are described in sufficient detail. However, consider providing more context or references for the analytical techniques used, particularly for readers who may be less familiar with these methods.

The section briefly mentions the statistical software used for analysis but lacks detail on specific statistical tests or procedures employed. Consider elaborating on the statistical methods used to analyze the data and determine significance.

Reference to Figure S1 is made but not included in the text. Ensure that all figures and tables mentioned in the text are properly labeled and referenced.

Ensure consistency in formatting, units of measurement, and terminology throughout the section to avoid confusion. Double-check numerical values and units for accuracy.

Provide information on replicates and controls to ensure the reproducibility of the experimental results. Clarify how triplicate experiments were conducted and averaged.

Results and Discussion

The section is well-organized and presents the results in a systematic manner. However, it could benefit from clearer subheadings to delineate different aspects of the results (e.g., soil physicochemical properties, soil nutrients availability, soil metals availability).

The interpretation of the results is generally clear, but consider providing more detailed explanations for some observations. For example, explain why certain types of microplastics (MPs) have different effects on soil pH, EC, and nutrient availability.

The section references previous studies to support the findings, which strengthens the discussion. However, provide more direct comparisons with previous research to highlight similarities or differences in results.

The figures and tables effectively complement the text and provide visual representations of the data. Ensure that all figures mentioned in the text are included, and provide clear captions for each figure/table to facilitate understanding.

Ensure consistency in reporting units of measurement and formatting of results throughout the section. Double-check numerical values and units for accuracy.

While the section presents results followed by discussions, aim for a more integrated approach where results are discussed in tandem with their implications. This will help readers understand the significance of the findings more effectively.

Ensure that all references cited in the text are included in the reference list, and vice versa. Check the formatting of citations to ensure consistency with the journal's style guidelines.

Provide more context for the findings within the broader scope of environmental science and soil research. Discuss the potential implications of the results for soil health, ecosystem functioning, and agricultural productivity.

While the section provides numerical data on soil properties and metal availability, consider including statistical analyses to support the significance of the findings. This could involve ANOVA, t-tests, or other relevant statistical tests to compare differences between treatment groups.

Address any limitations or potential sources of bias in the study design or methodology. Acknowledging limitations demonstrates transparency and helps readers interpret the results accurately.

Discuss the practical implications of the findings for environmental management or agricultural practices. Consider how the results could inform policies aimed at mitigating the impact of microplastics on soil health and metal contamination.

Suggest potential avenues for future research based on the study findings. Identify unanswered questions or areas requiring further investigation to advance understanding in this field.

Ensure that the language used is concise, precise, and appropriate for the target audience. Avoid overly technical jargon where possible and strive for clarity in conveying complex concepts.

Conclusions

Begin by succinctly summarizing the main findings of the study, emphasizing the significant changes observed in soil properties and heavy metal availability due to microplastic contamination.

Discuss the implications of the findings for soil health, environmental quality, and agricultural productivity. Emphasize the potential risks posed by microplastics and their impact on soil fertility and metal bioavailability.

Acknowledge the variation in effects observed among different types of microplastics and at different doses. Highlighting these variations adds depth to the conclusions and helps contextualize the results.

Suggest specific areas for future research, such as investigating the mechanisms underlying the interaction between microplastics and heavy metals, conducting plant experiments to assess their combined effects, and exploring the long-term implications for soil ecosystems.

Conclude with a call for action, emphasizing the importance of addressing microplastic pollution in soil and advocating for further research and policy initiatives to mitigate its impact on environmental and human health.

Best regards,

Reviewer #2: General comments:

In the manuscript “PONE-D-23-41961,” the authors evaluated the possible impact of microplastics (MPs) on soil physicochemical and chemical properties and toxic metals availability in contaminated and uncontaminated agricultural soils. The topic is relevant and timely. However, the study has important limitations that justify my recommendation against publication in a renowned journal such as PLOS ONE. The following items can help authors improve the manuscript for future publication.

Specific comments/suggestions:

1. Title – replace “physicochemical properties” >>> “physicochemical and chemical properties”

2. L21: replace “doses” >>> “concentrations”. Adjust this throughout the manuscript.

3. L23: define the acronyms or abbreviations presented in the abstract.

4. L60: replace “Ph” >>> “pH”.

5. L70: replace “MHs” >>> “HMs.” Check this throughout the manuscript.

6. Microplastics and MPs - When you first use a word that can be abbreviated, spell it out in full and show the abbreviation/acronym in parentheses immediately afterward. After that, you can stick to using the abbreviation/acronym.

7. L81-82: The justification for conducting the study should be improved. The absence of studies on a given topic does not constitute a good scientific justification. The goal of the first step of the manuscript is to get your audience's attention, to show them why your research matters, and to make them want to know more about your research. This step involves providing the reader with critical background or contextual information that introduces the topic area and indicates why the research is essential. Why is this research important? What real-life or everyday problem, issue, question, or context does the research relate to? What is the research ultimately trying to achieve? How can I grab the reader’s attention and concern? How can I state the problem or context of the research in terms that most people can report to? What possible negative repercussions would result from not solving this problem? What benefit does the research promise? How the intended outcomes of the study could be used to advance knowledge? These questions must be answered in the introduction of the manuscript! Furthermore, the hypotheses tested should be clearly presented at the end of the introduction.

8. L89-90: Were the plastic products new or aged? Clarify in the text.

9. L91-92: Authors must justify the choice of polymers tested. Although this is obvious, it must be clarified, considering that several other synthetic polymers could have been chosen.

10. L100-102: Authors must present the classification of the soil used. I recommend authors use the World Reference Base (WRB), i.e., the international standard for soil classification systems endorsed by the International Union of Soil Sciences.

11. L106: what is the reason for these seven days?

12. L113: Authors must clearly justify the choice of concentrations tested. The results of concentration-response experiments must always be interpreted in light of environmental concentrations.

13. L118: correct incorrect character in a sentence.

14. L116: What was the criterion for choosing the determined humidity? Throughout the 90 days of incubation, do the authors ensure this humidity was maintained unchanged? I'm not sure if the “paraffin” seal completely prevented evaporation. Clarify this in the text.

15. Were plastic cups used to conduct the experiment? Clarify this and how contradictory this can be in the context of the contamination of MPs.

16. L124: Anti-colorimetric technique? Confused! Clear that up!

17. L122-139: Authors must clarify the quality assurance/quality control (QA/QC) procedures adopted in the analyses.

18. Authors must analyze the data using appropriate statistics and explore it better, which can be achieved by consulting a statistics specialist. The findings reported by the authors were not adequately analyzed, which constitutes a significant limitation of the study. Furthermore, quantifying MPs in soils must confirm several speculations presented throughout the discussion. Without this quantification, it won't be easy to relate the variables investigated.

19. As long as I can remember, the rule has been that the title or caption for a table, along with the footnotes, to a table or figure, should provide enough information so that a reader can determine what the table or figure is showing without having to look for additional information in the text of the article. This means that all symbols and abbreviations must be defined in the table or the notes under the table, and the title must be very clear. This needs to be repeated separately for each table and figure on paper. Therefore, I recommend reforming the titles and captions of the figures/tables to make them self-explanatory.

20. Finally, careful proofreading in English should be carried out. Some sentences are difficult to understand. Thus, please write your text in good English (American or British usage is accepted, but not a mixture of these). Alternatively, authors may use the English Language Editing service available from Elsevier's Author Services.

6. PLOS authors have the option to publish the peer review history of their article (what does this mean?). If published, this will include your full peer review and any attached files.

Reviewer #1: **Yes: **Dr. Farzad Rassaei, PhD

Reviewer #2: No

---

## [Author Response · Author response to Decision Letter 0]

15 May 2024

Response to Reviewers

Manuscript ID: PONE-D-23-41961

Dear Editor, 

Thank you for giving us the opportunity to submit a revised draft of the manuscript "Short Communication: Evaluating the impacts of microplastics on agricultural soil physical, chemical properties and toxic metal availability: an emerging concern for sustainable agriculture" for the possibility of publication in the PLOS ONE. We appreciate the time and effort that you and the reviewers dedicated to providing feedback on our manuscript and are grateful for the insightful comments on and valuable improvements to our paper. We carefully considered the comments and tried our best to address every one of them. We hope the manuscript after careful revisions meet your high standards. The authors welcome further constructive comments if any. Below we provide the point-by-point responses. All modifications in the manuscript have been highlighted in yellow.

Sincerely yours, 

Md Simoon Nice

Department of Environmental Science and Technology

Jessore University of Science and Technology

Jessore-7408

Bangladesh

E-mail: simoon.nice.est.just@gmail.com

Response to Reviewer: 1

Author response: Thank you very much for reviewing our manuscript. We highly appreciate your valuable time and comments. We believe that your review comments significantly improve our manuscript quality. 

1. Abstract

The abstract effectively summarizes the main points of the study. However, some sentences could be further clarified and condensed for brevity without losing important information.

Provide more specific details about the methods used in the study, such as the experimental setup for soil incubation studies and the analytical techniques employed to measure soil characteristics and heavy metal availability.

Instead of listing all the changes in soil characteristics and heavy metal availability in a single sentence, consider breaking down the findings into separate sentences or bullet points to improve readability and comprehension.

Include specific quantitative results or ranges, such as percentage changes in soil properties and heavy metal concentrations, to provide readers with a clearer understanding of the magnitude of the observed effects.

Expand on the implications of the findings for soil health and environmental management. Discuss how the observed changes in soil properties and heavy metal availability due to MPs contamination might impact ecosystem functions and human health.

Ensure consistency in language use and grammar throughout the abstract. For example, check for proper tense usage and grammatical structure in each sentence.

The selected keywords effectively highlight the main topics of the study. However, consider adding additional keywords related to soil pollution, environmental impact assessment, or plastic pollution to improve the discoverability of the manuscript.

Provide a concise summary of the key conclusions drawn from the study's findings, emphasizing the significance of the research and potential avenues for future investigation.

Author response: We highly appreciate your valuable comments, thank you. We revised it in the manuscript according to your suggestion made it in the manuscript. Specific details about the method are included in [L18-23]. We have also added soil pollution as a keyword. Abstract writing pattern follow the given article. Please consider this issue.

Feng, X., Wang, Q., Sun, Y., Zhang, S., & Wang, F. (2022). Microplastics change soil properties, heavy metal availability and bacterial community in a Pb-Zn-contaminated soil. Journal of Hazardous materials, 424, 127364.

de Souza Machado, A. A., Lau, C. W., Kloas, W., Bergmann, J., Bachelier, J. B., Faltin, E., ... & Rillig, M. C. (2019). Microplastics can change soil properties and affect plant performance. Environmental science & technology, 53(10), 6044-6052.

Yu, H., Hou, J., Dang, Q., Cui, D., Xi, B., & Tan, W. (2020). Decrease in bioavailability of soil heavy metals caused by the presence of microplastics varies across aggregate levels. Journal of hazardous materials, 395, 122690.

de Souza Machado, A. A., Lau, C. W., Till, J., Kloas, W., Lehmann, A., Becker, R., & Rillig, M. C. (2018). Impacts of microplastics on the soil biophysical environment. Environmental science & technology, 52(17), 9656-9665.

2. Introduction

The Introduction effectively introduces the topic of microplastic pollution in soil and highlights its significance for soil health and ecosystem functioning. The structure of the Introduction is logical, with clear transitions between different aspects of the topic.

The Introduction provides a thorough review of relevant literature, citing key studies to support the discussion on the sources, distribution, and potential impacts of microplastics in soil ecosystems. However, ensure that all cited studies are accurately referenced and consider including more recent references to reflect the latest research in the field.

The objectives of the study are clearly stated, indicating the focus on assessing the impact of MPs on soil properties and heavy metal availability in both contaminated and uncontaminated agricultural soils. Consider briefly mentioning the specific MPs types and doses tested in the study to provide additional context for readers.

The Introduction effectively identifies knowledge gaps in the current understanding of microplastic pollution in terrestrial environments, emphasizing the need for research on MPs in agricultural soils. This sets the stage for the study's contribution to addressing this gap.

Ensure smooth transitions between paragraphs and sections to maintain the flow of the Introduction. Consider revising sentence structures for improved clarity and readability, avoiding overly complex or convoluted phrasing.

Conclude the Introduction by emphasizing the importance of the study's findings for informing soil management practices, environmental policy, and future research directions in the field of microplastic pollution in terrestrial ecosystems.

The review of intermittent irrigation techniques is good but citing key studies would strengthen the evaluation (e.g. "Recent work like doi.org/10.1002/ep.14301, https://doi.org/10.1080/15320383.2023.2258413, https://doi.org/10.1080/15226514.2023.2250464, doi.org/10.1002/ep.14230, doi.org/10.1002/ep.14217, https://doi.org/10.1007/s10333-022-00915-5

Author response: Thank you for valuable suggestion. We have reviewed the latest articles and cited those (Ref no 18-24) studies in the manuscript [L63-64].

3. Materials and Methods" section needs significant revisions for clarity and completeness:

The section effectively outlines the steps involved in microplastic preparation, soil preparation, experimental setup, and analysis methods. However, some sentences are lengthy and complex, which may affect readability. Consider breaking down long sentences into shorter ones for clarity.

The description of microplastic preparation is clear and provides sufficient detail on the sources of different types of microplastics and the cleaning process. Ensure consistency in terminology (e.g., "plastics items" vs. "plastic products") for clarity.

The description of soil collection, preparation, and experimental setup is well-detailed, including information on soil properties and treatment groups. Consider providing more information on the rationale for choosing specific microplastic types and doses for the experiment.

The methods for analyzing soil properties, including pH, nutrient content, organic carbon, and heavy metals, are described in sufficient detail. However, consider providing more context or references for the analytical techniques used, particularly for readers who may be less familiar with these methods.

The section briefly mentions the statistical software used for analysis but lacks detail on specific statistical tests or procedures employed. Consider elaborating on the statistical methods used to analyze the data and determine significance.

Reference to Figure S1 is made but not included in the text. Ensure that all figures and tables mentioned in the text are properly labeled and referenced.

Ensure consistency in formatting, units of measurement, and terminology throughout the section to avoid confusion. Double-check numerical values and units for accuracy.

Provide information on replicates and controls to ensure the reproducibility of the experimental results. Clarify how triplicate experiments were conducted and averaged.

Author response: Thank you for pointing this out. We have according made it in the manuscript. We have changed the long complex sentences into shorter and simple one to increase the readability. "Plastics items" are replaced by "plastic products" to maintain consistency in terminology for clarity. More information on the rationale for choosing specific microplastic types and doses for the experiment are given in [L119-124], with references. The methods for analyzing soil properties are added in details with references as well in [L132-161]. Specific statistical tests and procedure are mentioned in [L163-166]. We have mentioned all the tables and figures accordingly as well. Details on replication is given in [L158-161].

4. Results and Discussion

The section is well-organized and presents the results in a systematic manner. However, it could benefit from clearer subheadings to delineate different aspects of the results (e.g., soil physicochemical properties, soil nutrients availability, soil metals availability).

The interpretation of the results is generally clear, but consider providing more detailed explanations for some observations. For example, explain why certain types of microplastics (MPs) have different effects on soil pH, EC, and nutrient availability.

The section references previous studies to support the findings, which strengthens the discussion. However, provide more direct comparisons with previous research to highlight similarities or differences in results.

The figures and tables effectively complement the text and provide visual representations of the data. Ensure that all figures mentioned in the text are included, and provide clear captions for each figure/table to facilitate understanding.

Ensure consistency in reporting units of measurement and formatting of results throughout the section. Double-check numerical values and units for accuracy.

While the section presents results followed by discussions, aim for a more integrated approach where results are discussed in tandem with their implications. This will help readers understand the significance of the findings more effectively.

Ensure that all references cited in the text are included in the reference list, and vice versa. Check the formatting of citations to ensure consistency with the journal's style guidelines.

Provide more context for the findings within the broader scope of environmental science and soil research. Discuss the potential implications of the results for soil health, ecosystem functioning, and agricultural productivity.

While the section provides numerical data on soil properties and metal availability, consider including statistical analyses to support the significance of the findings. This could involve ANOVA, t-tests, or other relevant statistical tests to compare differences between treatment groups.

Address any limitations or potential sources of bias in the study design or methodology. Acknowledging limitations demonstrates transparency and helps readers interpret the results accurately.

Discuss the practical implications of the findings for environmental management or agricultural practices. Consider how the results could inform policies aimed at mitigating the impact of microplastics on soil health and metal contamination.

Suggest potential avenues for future research based on the study findings. Identify unanswered questions or areas requiring further investigation to advance understanding in this field.

Ensure that the language used is concise, precise, and appropriate for the target audience. Avoid overly technical jargon where possible and strive for clarity in conveying complex concepts.

Author response: We highly appreciate the reviewer's comment. The interpretation of the results has been supported by providing more detailed explanations and justifications for the observations [L175-177; L195-200; L201-209; L210-215]. We have also double-checked the values and unit’s accuracy to ensure the consistency in reporting. We performed ANOVA test to support the significance, those are shown in Fig 1, Fig 2 and Table 1 [L220-222; L255-256; L258-259; L279-280; L283-284]. The limitations of the study are also addressed in [L285-292].

5. Conclusions

Begin by succinctly summarizing the main findings of the study, emphasizing the significant changes observed in soil properties and heavy metal availability due to microplastic contamination.

Discuss the implications of the findings for soil health, environmental quality, and agricultural productivity. Emphasize the potential risks posed by microplastics and their impact on soil fertility and metal bioavailability.

Acknowledge the variation in effects observed among different types of microplastics and at different doses. Highlighting these variations adds depth to the conclusions and helps contextualize the results.

Suggest specific areas for future research, such as investigating the mechanisms underlying the interaction between microplastics and heavy metals, conducting plant experiments to assess their combined effects, and exploring the long-term implications for soil ecosystems.

Conclude with a call for action, emphasizing the importance of addressing microplastic pollution in soil and advocating for further research and policy initiatives to mitigate its impact on environmental and human health.

Author response: Thank you for pointing this out. We have made the changes in the manuscript. The conclusion summarized the main finding of the study at the beginning. Effect due to the dose variation is also acknowledged in the manuscript [L302-303]. Future research scope is also outlined in [L303-307].

Response to Reviewer: 2

Reviewer #2: In the manuscript “PONE-D-23-41961,” the authors evaluated the possible impact of microplastics (MPs) on soil physicochemical and chemical properties and toxic metals availability in contaminated and uncontaminated agricultural soils. The topic is relevant and timely. However, the study has important limitations that justify my recommendation against publication in a renowned journal such as PLOS ONE. The following items can help authors improve the manuscript for future publication.

Author response: Thank you very much for reviewing our manuscript. We highly appreciate your valuable time and comments. We believe that your review comments significantly improve our manuscript quality. 

1. Title – replace “physicochemical properties” >>> “physicochemical and chemical properties”

Author response: Thank you for pointing this out. We have made it accordingly in the title.

2. L21: replace “doses” >>> “concentrations”. Adjust this throughout the manuscript.

Author response: Thank you for your suggestion. We have replaced doses with concentrations throughout the manuscript. 

3. L23: define the acronyms or abbreviations presented in the abstract.

Author response: Thank you for valuable suggestion. We have accordingly made it in the manuscript.

4. L60: replace “Ph” >>> “pH”.

Author response: Thank you for pointing this out. It was a typing mistake. We have changed it properly. [L-62]

5. L70: replace “MHs” >>> “HMs.” Check this throughout the manuscript.

Author response: Thank you for pointing this out. It was a typing mistake. We have changed it properly throughout the manuscript. [L-73]

6. Microplastics and MPs - When you first use a word that can be abbreviated, spell it out in full and show the abbreviation/acronym in parentheses immediately afterward. After that, you can stick to using the abbreviation/acronym.

Author response: Thank you for your valuable suggestion. We have made the adjustment in the manuscript accordingly.

7. L81-82: The justification for conducting the study should be improved. The absence of studies on a giv

---

## [Decision Letter · Decision Letter 1]

20 May 2024

Short Communication: Evaluating the impacts of microplastics on agricultural soil physicochemical properties and toxic metal availability: an emerging concern for sustainable agriculture

PONE-D-23-41961R1

Dear Md. Simoon Nice,

We’re pleased to inform you that your manuscript has been judged scientifically suitable for publication and accepted for publication. 

Kind regards,

Babar Iqbal, PhD

Academic Editor

PLOS ONE

Additional Editor Comments (optional):

The reviewer is satisfied with the revision and recommend accepting the manuscript in its current form.

---

## [Editor Report · Acceptance letter]

14 Jun 2024

PONE-D-23-41961R1 

PLOS ONE

Dear Dr. Nice, 

I'm pleased to inform you that your manuscript has been deemed suitable for publication in PLOS ONE. Congratulations! Your manuscript is now being handed over to our production team.

Kind regards, 

on behalf of

Dr. Babar Iqbal 

Academic Editor

PLOS ONE